# Geometric IB: Improving Information Bottleneck with Geometry-Aware Compression on Statistical Manifolds

## Abstract

We revisit the Information Bottleneck (IB) through the lens of information geometry and propose a Geometric Information Bottleneck (G-IB) that dispenses with direct mutual information (MI) estimation. We show that mutual information $I(X; Z)$ and $I(Z; Y)$ admit exact projection forms as minimal Kullback–Leibler (KL) distances from the joint distributions to their respective independence manifolds. Guided by this view, G-IB controls information compression with two complementary terms: (i) a distribution-level Fisher–Rao (FR) discrepancy, which matches KL to second order and is reparameterization-invariant; and (ii) a geometry-level Jacobian–Frobenius (JF) term that provides a local capacity-type upper bound on $I_\phi(Z; X)$ by penalizing pullback volume expansion of the encoder. We further derive a natural-gradient optimizer consistent with the FR metric and prove that the standard additive natural-gradient step is first-order equivalent to the geodesic update. We conducted extensive experiments and observed that the G-IB achieves a better trade-off between prediction accuracy and compression ratio in the information plane than the mainstream IB baselines on popular datasets. G-IB offers a principled and scalable alternative that unifies distributional and geometric regularization under a single bottleneck multiplier, improving invariance and optimization stability. The source code of G-IB is released at https://anonymous.4open.science/r/G-IB-0569.

## 1 Introduction

The Information Bottleneck (IB) principle (Tishby et al., 2000) casts representation learning as extracting a representation $Z$ from $X$ that preserves only what is useful for predicting $Y$. Concretely, one seeks an encoder $q_\phi(z \mid x)$ such that $Z$ carries as much information about $Y$ as possible while remaining maximally compressed with respect to $X$, which can be formulated as:

$$\max_\phi \ I_\phi(Z; Y) \quad \text{s.t.} \quad I_\phi(X; Z) \leq R, \tag{1}$$

where $I(\cdot; \cdot)$ denotes the mutual information, $\phi$ are the parameters of the encoder, and $R$ sets the compression budget. Here, $I_\phi(\cdot; \cdot)$ is computed under the data distribution $p(x, y)$ and the encoder $q_\phi(z \mid x)$. To address this constrained optimization problem, the IB method ((Tishby et al., 2000; Alemi et al., 2016)) introduces a positive Lagrange multiplier $\beta$, transforming the problem into

$$\min_\phi -I_\phi(Z; Y) + \beta I_\phi(X; Z), \tag{2}$$

where $\beta \geq 0$ (the "bottleneck" parameter) balances predictive sufficiency against compression.

The IB principle is appealing because it formalizes what constitutes a useful representation via a fundamental balance between compression and predictive sufficiency (Alemi et al., 2016; Tishby et al., 2000; Wu et al., 2020a). Thus, IB and a wide range of variants (Wan et al., 2021; Yang et al., 2025; Yu et al., 2024; Zhai & Zhang, 2022) have been adopted across diverse applications, including image segmentation (Xu et al., 2024), domain generalization (Li et al., 2022), semantic communication (Xie et al., 2023; Wang et al., 2024), and privacy compression (Dubois et al., 2021; Razeghi et al., 2023). Moreover, prior work (Shwartz-Ziv & Tishby, 2017) suggests that IB provides a principled lens for interpreting certain training dynamics of deep neural networks and unveil universal attrition to interpret vision transformers (Hong et al., 2025).

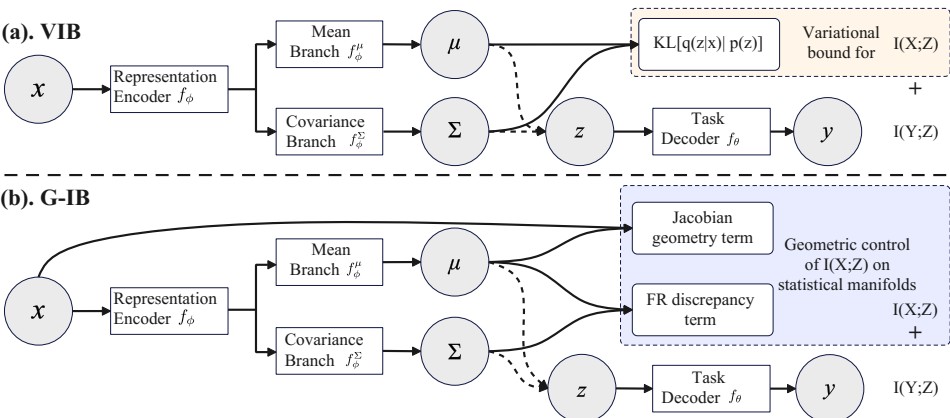

Figure 1: Comparison of VIB and G-IB. Both models parametrize the encoder as $q_\phi(z \mid x) = \mathcal{N}(\mu(x), \mathrm{diag}(\sigma^2(x)))$ by a network $f_\phi$ and use a task decoder $p_\theta(y \mid z)$ to increase $I(Z; Y)$ by a network $f_\theta$. (a) VIB: compression is enforced by the variational upper bound. (b) G-IB: replaces explicit MI estimation with two geometry-aware penalties computed deterministically on statistical manifolds: a Fisher–Rao quadratic proxy $\mathcal{L}_{\mathrm{FR}}$ and a Jacobian-Frobenius term $\mathcal{L}_{\mathrm{JF}}$. Solid arrows denote deterministic mappings; dashed arrows indicate reparameterized sampling $z = \mu + \sigma \odot \varepsilon$.

**Research Gap.** Despite this success, most practical IB implementations optimize Euclidean surrogates of mutual information (MI), e.g., variational bounds as in VIB (Alemi et al., 2016) or neural MI estimators such as MINE (Belghazi et al., 2018). These surrogates disregard the statistical-manifold geometry of the encoder or posterior family and thus offer no explicit geometric guarantees (e.g., reparameterization invariance, curvature-aware regularization) over $I(X; Z)$. Recent extensions, such as structure IB (Yang et al., 2025; Hu et al., 2024), try to extract the structure information from the input. However, they still rely on Euclidean MI proxies, often degrading accuracy in the strong-compression regime and making results highly sensitive to $\beta$ and estimator hyperparameters.

**Research Question.** This motivates the following research question: "Can we design a geometry-aware IB that operates on statistical manifolds and provides stable, principled control of representation compression?"

In this paper we introduce the Geometric Information Bottleneck (G-IB), which reframes IB through the lens of statistical–manifold geometry. We first establish exact projection characterizations: both $I(X; Z)$ and $I(Z; Y)$ can be written as minimal Kullback–Leibler (KL) distances from the corresponding joint distributions to their independence submanifolds. Then, we design the G-IB method, which regulates compression via two complementary components: (i) a *distribution-level* Fisher–Rao (FR) discrepancy that agrees with KL to second order and is invariant under smooth reparameterizations of $z$; and (ii) a *geometry-level* Jacobian–Frobenius (JF) penalty that yields a local capacity–type upper bound on $I_\phi(Z; X)$ by discouraging pullback volume expansion of the encoder. Finally, we derive an optimizer with respect to the Fisher–Rao (FR) metric whose update direction is the natural gradient, and we prove first-order equivalence with the geodesic update. To summarize, we make the following contributions:

- **Geometric Reformulation of IB.** We show that both $I(Z; X)$ and $I(Z; Y)$ admit exact projection forms as minimal KL distances from the joint distributions to their respective independence manifolds, clarifying the geometric structure underlying the IB principle.
- **A G-IB Solution.** We propose G-IB, which controls compression via two complementary terms: (i) a distribution-level Fisher–Rao (FR) discrepancy and (ii) a geometry-level Jacobian–Frobenius (JF) penalty. We also derive a natural-gradient optimizer consistent with the FR metric and prove that: the standard additive natural-gradient step is first-order equivalent to the geodesic (exponential-map) update.
- **Empirical Validation.** We conducted extensive experiments to compare with representative benchmarks. G-IB attains favorable accuracy–compression trade-offs in the information plane relative to the state-of-the-art IB baselines, with improved robustness in strong compression regimes.

As the page limitation, we provide the **Related Work** Section in Appendix B.

## 2 PROBLEM STATEMENT FROM A GEOMETRIC VIEW

Let $\mathcal{P}$ be the statistical manifold of all joint distributions over $(X, Z)$ and let the independence manifold be $\mathcal{I}_{XZ} = \{ q(x)r(z) \}$. In exponential (e-) coordinates $\mathcal{I}_{XZ}$ is a e-flat submanifold (Amari & Nagaoka, 2000; Amari, 2016). For any $q$ and $r$, we have the information-geometric Pythagorean identity (Amari & Nagaoka, 2000)

$$\mathrm{KL}\big(p_\phi(x,z)\,\|\,q(x)r(z)\big) = \underbrace{\mathrm{KL}\big(p_\phi(x,z)\,\|\,p(x)p_\phi(z)\big)}_{=\,I_\phi(X;Z)} + \underbrace{\mathrm{KL}\big(p(x)p_\phi(z)\,\|\,q(x)r(z)\big)}_{\geq 0}, \quad (3)$$

whence $I_\phi(Z;X) = \min_{q,r} \mathrm{KL}\big(p_\phi(x,z)\,\|\,q(x)r(z)\big)$ with minimizer $(q,r) = (p(x), p_\phi(z))$. An identical relation holds for $(Y, Z)$ by replacing $x$ with $y$, $q$ with $q'$, and $r$ with $r'$:

$$\mathrm{KL}\big(p_\phi(y,z)\,\|\,q'(y)r'(z)\big) = I_\phi(Z;Y) + \mathrm{KL}\big(p(y)p_\phi(z)\,\|\,q'(y)r'(z)\big). \quad (4)$$

We assume absolute continuity so that all KL terms are finite and Fubini's theorem (Kallenberg, 1997) applies; in particular $p_\phi(x,z) \ll q(x)r(z)$ and $p_\phi(y,z) \ll q'(y)r'(z)$ for candidate product measures. We provide the detailed proof of Eq. (3) in Appendix C, and Eq. (4) can also be proved in the same way. The Information Bottleneck loss (Tishby et al., 2000), $\mathcal{L}_{\mathrm{IB}}(\phi) = \beta\, I_\phi(Z;X) - I_\phi(Z;Y)$, can thus be written exactly as

$$\mathcal{L}_{\mathrm{IB}}(\phi) = \beta \min_{q,r} \mathrm{KL}\big(p_\phi(x,z)\,\|\,q(x)r(z)\big) - \min_{q',r'} \mathrm{KL}\big(p_\phi(y,z)\,\|\,q'(y)r'(z)\big), \quad (5)$$

where the inner minima are achieved at $(q,r) = (p(x), p_\phi(z))$ and $(q',r') = (p(y), p_\phi(z))$. Optimizing Eq. (5) over $\phi$ is therefore equivalent to:

- *push $p_\phi(x,z)$ toward the independence manifold $\mathcal{I}_{XZ}$ by minimizing $\beta\, \mathrm{KL}\big(p_\phi(x,z)\,\|\,p(x)p_\phi(z)\big)$;*

- *pull $p_\phi(y,z)$ away from its independence manifold $\mathcal{I}_{YZ} = \{q'(y)r'(z)\}$ by maximizing $\mathrm{KL}\big(p_\phi(y,z)\,\|\,p(y)p_\phi(z)\big)$.*

Although the projection-based formulation is exact, directly optimizing it is computationally challenging due to the need to evaluate KL projections and encoder marginals at scale. We introduces an approximate solution that replaces explicit MI estimation with geometry-derived surrogates.

## 3 GEOMETRIC INFORMATION BOTTLENECK METHOD

**Overview.** As introduced in the above section, we recast the IB objective through information geometry: the mutual information $I_\phi(X;Z)$ and $I_\phi(Z;Y)$ are the minimal KL distances from the joint distributions $p_\phi(x,z)$ and $p_\phi(y,z)$ to their respective independence manifolds, turning the IB Lagrangian into a difference of projection distances. Based on this, our Geometric IB (G-IB) controls compression with two complementary terms: (i) a *distribution-level* Fisher–Rao (FR) discrepancy and (ii) a *geometry-level* Jacobian–Frobenius (JF) term. During training, we propose a natural gradient descent method that combines the geometric information to achieve a better optimization effect.

### 3.1 DISTRIBUTION PROXY VIA THE FISHER–RAO (FR) QUADRATIC

We approximate the conditional–marginal divergence that defines the compression term by the local second-order FR metric:

$$I_\phi(Z;X) = \mathbb{E}_{p(x)} D_{\mathrm{KL}}\big(q_\phi(z|x)\,\|\,p_\phi(z)\big) \approx \tfrac{1}{2}\mathbb{E}_{p(x)} d_{\mathrm{FR}}\big(q_\phi(z|x), r(z)\big)^2, \quad (6)$$

where $r(z)$ is a reference marginal (e.g., a standard normal or a learned prior), and $d_{\mathrm{FR}}(\cdot,\cdot)$ denotes the Fisher–Rao geodesic distance between distributions, i.e., the Riemannian distance induced by the Fisher information metric. The approximation follows from the local equivalence for smooth parametric families $\{p_\theta\}$, for $\theta' = \theta + \Delta$ with $\|\Delta\|$ small:

$$D_{\mathrm{KL}}(p_{\theta'}\,\|\,p_\theta) = \tfrac{1}{2}\Delta^\top F(\theta)\,\Delta + o(\|\Delta\|^2) = \tfrac{1}{2} d_{\mathrm{FR}}\big(p_{\theta'}, p_\theta\big)^2 + o(\|\Delta\|^2), \quad (7)$$

with $F(\theta)$ the Fisher information. Replacing $p_{\theta'}$ by $q_\phi(z|x)$ and $p_\theta$ by $r(z)$ in Eq. (7) yields Eq. (6). We provide the proof in Appendix D.

**Diagonal Gaussian Example.** If $q_\phi(z|x) = \mathcal{N}\big(\mu_\phi(x), \mathrm{diag}(\sigma_\phi^2(x))\big)$ and $r(z) = \mathcal{N}(0, I)$, the exact KL is

$$D_{\mathrm{KL}}\big(q_\phi(z|x) \,\|\, \mathcal{N}(0, I)\big) = \tfrac{1}{2} \sum_{j=1}^{d_z} \Big( \mu_j(x)^2 + \sigma_j(x)^2 - \log \sigma_j(x)^2 - 1 \Big). \tag{8}$$

Expanding Eq. (8) at $\mu = 0$, $\sigma^2 = 1$ gives

$$D_{\mathrm{KL}}\big(q_\phi \| \mathcal{N}(0, I)\big) = \tfrac{1}{2}\|\mu\|_2^2 + \tfrac{1}{4}\|\log \sigma^2\|_2^2 + O(\|\log \sigma^2\|_2^3), \tag{9}$$

which equals $\tfrac{1}{2} d_{\mathrm{FR}}\big(q_\phi, \mathcal{N}(0, I)\big)^2$ up to second order. In practice, when $r = \mathcal{N}(0, I)$ we can optimize the closed-form KL in Eq. (8): $\mathcal{L}_{\mathrm{KL}}(\phi) = \widehat{\mathbb{E}}_x D_{\mathrm{KL}}\big(q_\phi(z|x) \| r(z)\big)$. If $r$ is learned/complex (e.g., VampPrior/flow), we may optimize the FR proxy

$$\mathcal{L}_{\mathrm{FR}}(\phi) = \tfrac{1}{2} \widehat{\mathbb{E}}_x \, d_{\mathrm{FR}}\big(q_\phi(z|x), r(z)\big)^2, \tag{10}$$

and optionally monitor $\widehat{I}_{XZ}^{\mathrm{KL}} = \widehat{\mathbb{E}}_x D_{\mathrm{KL}}\big(q_\phi(z|x) \| \widehat{p}_\phi(z)\big)$ to gauge tightness.

### 3.2 GEOMETRIC BOUND VIA THE JACOBIAN–FROBENIUS TERM

Assume a reparameterized encoder $z = f_\phi(x) + \varepsilon$, where $\varepsilon \sim \mathcal{N}\big(0, \Sigma(x)\big)$, and denote the Jacobian $J_f(x) = \partial f_\phi(x)/\partial x$. The pullback metric on the input manifold is $g_x = J_f(x)^\top \Sigma(x)^{-1} J_f(x)$.

**Local Capacity-type Upper Bound.** Linearizing $f_\phi$ around $x$ and letting $C_x$ be the local input covariance, a Gaussian channel upper bound yields

$$I_\phi(Z; X) \le \tfrac{1}{2} \mathbb{E}_{p(x)} \left[ \log \det \Big( I + \Sigma(x)^{-\frac{1}{2}} J_f(x) \, C_x \, J_f(x)^\top \Sigma(x)^{-\frac{1}{2}} \Big) \right]. \tag{11}$$

Under a unit local energy constraint $C_x \preceq I$ (Loewner order) and the monotonicity of $\log \det$ on PSD cone $\mathbb{S}_+^d$, we obtain the pointwise bound

$$I_\phi(Z; X) \le \tfrac{1}{2} \mathbb{E}_{p(x)} \left[ \log \det \Big( I + \Sigma(x)^{-\frac{1}{2}} J_f(x) J_f(x)^\top \Sigma(x)^{-\frac{1}{2}} \Big) \right] \tag{12}$$

$$= \tfrac{1}{2} \mathbb{E}_{p(x)} \left[ \log \det \Big( I + J_f(x)^\top \Sigma(x)^{-1} J_f(x) \Big) \right], \tag{13}$$

$$\le \tfrac{1}{2} \mathbb{E}_{p(x)} \, \mathrm{Tr}\big( \Sigma(x)^{-1} J_f(x) J_f(x)^\top \big) = \tfrac{1}{2} \mathbb{E}_{p(x)} \big\| \Sigma(x)^{-\frac{1}{2}} J_f(x) \big\|_F^2 =: \tfrac{1}{2} \mathrm{JF}(\phi). \tag{14}$$

*Proof sketch.* Since $Z$ depends on $X$ only through $(f_\phi(X), \Sigma(X))$, we have the Markov chain $X \to (f_\phi(X), \Sigma(X)) \to Z$ and thus $I(X; Z) = I\big((f_\phi(X), \Sigma(X)); Z\big)$. Conditioning on $x$ and linearizing $f_\phi$ at $x$ while holding $\Sigma(x)$ fixed locally yields a (local) linear Gaussian channel with gain $J_f(x)$ and noise covariance $\Sigma(x)$. Under a unit local energy constraint on the input covariance $C_x \preceq I$, the Gaussian maximizes entropy for fixed covariance, giving the log-det bound in Eq. (11). Using $\det(I + AB) = \det(I + BA)$ gives Eq. (13), and applying $\log \det(I + A) \le \mathrm{Tr}(A)$ for $A \succeq 0$ yields Eq. (14). $\qquad\square$

**Unbiased Hutchinson Estimator.** The trace in Eq. (14) can be estimated without forming explicit Jacobians. For any $v \sim \mathcal{N}(0, I_{d_x})$,

$$\mathbb{E}_v \big[ \|\Sigma(x)^{-1/2} J_f(x) v\|_2^2 \big] = \mathrm{Tr}\big( \Sigma(x)^{-1} J_f(x) J_f(x)^\top \big). \tag{15}$$

With $S$ i.i.d. probe vectors $\{v_s\}_{s=1}^S$, define the per-sample estimator

$$\widehat{\mathrm{JF}}(x) := \frac{1}{S} \sum_{s=1}^S \big\| \Sigma(x)^{-1/2} J_f(x) \, v_s \big\|_2^2, \quad \text{so that} \quad \mathbb{E}_v[\widehat{\mathrm{JF}}(x)] = \mathrm{Tr}\big( \Sigma^{-1} J_f(x) J_f(x)^\top \big). \tag{16}$$

The training objective is the batch average

$$\mathcal{L}_{\mathrm{JF}}(\phi) := \widehat{\mathbb{E}}_{x \sim \mathrm{batch}} \big[ \widehat{\mathrm{JF}}(x) \big]. \tag{17}$$

In automatic differentiation frameworks, compute $J_f(x)v_s$ via Jacobian–vector products (JVP), costing $O(S)$ forward-mode calls per $x$; $S = 1$ or $2$ is typically sufficient.

**Isotropic/diagonal noise.** If $\Sigma(x) = \sigma(x)^2 I$, then

$$I_\phi(Z; X) \ \le \ \tfrac{1}{2} \, \mathbb{E}_{p(x)} \log \det \Big( I + \tfrac{1}{\sigma(x)^2} J_f J_f^\top \Big) \ \le \ \tfrac{1}{2} \, \widehat{\mathbb{E}}_x \, \frac{\|J_f(x)\|_F^2}{\sigma(x)^2}. \tag{18}$$

With a scalar floor $\sigma_{\min}^2 > 0$, this yields a simple, stable surrogate.

*Geometric meaning.* Because $g_x = J_f^\top \Sigma^{-1} J_f$, we have $\mathrm{Tr}(g_x) = \|\Sigma(x)^{-\frac{1}{2}} J_f(x)\|_F^2$, the direction-averaged local stretch (Dirichlet energy density under the $\Sigma^{-1}$ metric). Minimizing the JF term therefore controls average geodesic-length distortion, providing a principled compression surrogate for $I_\phi(Z; X)$.

## 3.3 Natural-Gradient Optimization for G-IB

Building on the above, we formulate the G-IB objective as

$$\mathcal{L}_{\mathrm{G\text{-}IB}}(\phi, \theta) \ = \ \underbrace{\mathbb{E}_{p(x,y)} \mathbb{E}_{q_\phi(z|x)} \big[ -\log p_\theta(y \mid z) \big]}_{\uparrow I_\phi(Z;Y)} + \underbrace{\beta \Big( \widehat{\mathcal{L}}_{\mathrm{FR}}(\phi) + \widehat{\mathcal{L}}_{\mathrm{JF}}(\phi) \Big)}_{\downarrow I_\phi(Z;X)}, \tag{19}$$

where $\beta \ge 0$ is the bottleneck multiplier. Minimizing $\mathbb{E}_{p(x,y)} \mathbb{E}_{q_\phi(z|x)} \big[ -\log p_\theta(y \mid z) \big]$ reduces $H(Y \mid Z)$ and increases $I_\phi(Z; Y)$ (since $H(Y)$ is fixed); the FR and JF terms jointly penalize $I_\phi(Z; X)$.

**Natural Gradient on the Encoder.** Viewing $\{q_\phi(z \mid x)\}_\phi$ as a statistical manifold endowed with the Fisher–Rao metric, the natural gradient of any scalar objective $\mathcal{J}(\phi)$ is

$$\widetilde{\nabla}_\phi \mathcal{J} \ = \ F_\phi^{-1} \nabla_\phi \mathcal{J}, \qquad F_\phi \ := \ \mathbb{E}_{p(x)} \mathbb{E}_{q_\phi(z|x)} \big[ \nabla_\phi \log q_\phi(z \mid x) \, \nabla_\phi \log q_\phi(z \mid x)^\top \big], \tag{20}$$

where the matrix $F_\phi$ in Eq. (20) is the Fisher–Rao metric tensor. We update the encoder by preconditioning the Euclidean gradient of the full objective:

$$\phi_{t+1} \ = \ \phi_t - \eta_\phi \, F_{\phi_t}^{-1} \Big( \nabla_\phi \, \mathbb{E}_{x,y,z}[-\log p_\theta(y \mid z)] + \beta \big[ \nabla_\phi \mathcal{L}_{\mathrm{FR}} + \nabla_\phi \mathcal{L}_{\mathrm{JF}} \big] \Big), \tag{21}$$

which is first-order invariant under smooth reparameterizations of $\phi$ and couples the distribution-level FR and geometry-level JF signals through a single preconditioner $F_\phi$. We can also prove that the additive natural-gradient step $\phi^+ = \phi - \eta F_\phi^{-1} \nabla_\phi \mathcal{J}$ is a first-order approximation to the geodesic update in geometry.

**Proposition 1** (Natural gradient equals the Riemannian gradient)**.** *Let $\mathcal{M} = \{p_\phi : \phi \in \Theta \subset \mathbb{R}^d\}$ be a regular statistical manifold endowed with the Fisher–Rao metric $g_\phi(u, v) := u^\top F_\phi v$, where $F_\phi = \mathbb{E}_{p(x)} \mathbb{E}_{q_\phi(z|x)} [\nabla_\phi \log q_\phi \, \nabla_\phi \log q_\phi^\top]$. For a scalar objective $\mathcal{J} : \mathcal{M} \to \mathbb{R}$, its Riemannian gradient at $\phi$ satisfies*

$$\mathrm{grad}\, \mathcal{J}(\phi) \ = \ F_\phi^{-1} \nabla_\phi \mathcal{J}, \tag{22}$$

*i.e., the natural gradient $\widetilde{\nabla}_\phi \mathcal{J} := F_\phi^{-1} \nabla_\phi \mathcal{J}$ is exactly the Riemannian gradient on $(\mathcal{M}, g)$.*

See proof in Appendix E.

**Proposition 2** (Steepest descent under the Fisher–Rao metric)**.** *Let $(\mathcal{M}, g)$ be endowed with the Fisher–Rao metric $g_\phi(u, v) = u^\top F_\phi v$ and let $\mathcal{J}$ be smooth. The direction of steepest descent per unit FR length solves*

$$\min_{\|v\|_{g_\phi} \le 1} \ \mathrm{D}\mathcal{J}(\phi)[v], \tag{23}$$

*and the (unit-norm) optimizer is $v_\star = -\frac{\mathrm{grad}\, \mathcal{J}(\phi)}{\|\mathrm{grad}\, \mathcal{J}(\phi)\|_{g_\phi}}$, with the convention $v_\star = 0$ if $\mathrm{grad}\, \mathcal{J}(\phi) = 0$. In particular, by Proposition 1, its direction coincides with the negative natural gradient: $-\mathrm{grad}\, \mathcal{J}(\phi) \equiv -F_\phi^{-1} \nabla_\phi \mathcal{J}$.*

*Proof sketch.* By the Riemannian gradient definition in (Amari & Nagaoka, 2000), $D\mathcal{J}(\phi)[v] = \langle \operatorname{grad} \mathcal{J}(\phi), v \rangle_{g_\phi}$. Cauchy–Schwarz gives $\langle \operatorname{grad} \mathcal{J}, v \rangle_{g_\phi} \geq -\|\operatorname{grad} \mathcal{J}\|_{g_\phi} \|v\|_{g_\phi} \geq -\|\operatorname{grad} \mathcal{J}\|_{g_\phi}$, with equality iff $v$ is collinear with $-\operatorname{grad} \mathcal{J}$ and $\|v\|_{g_\phi} = 1$. $\qquad\square$

By Proposition 2, the steepest descent direction per unit FR length is $-\operatorname{grad} \mathcal{J}(\phi)$. We thus update along this direction using the exponential map.

**Theorem 1** (Geodesic update via the exponential map). *Let* $\operatorname{Exp}_\phi : T_\phi \mathcal{M} \to \mathcal{M}$ *be the Riemannian exponential map of the FR metric. The discrete update*

$$\phi^+ = \operatorname{Exp}_\phi\big(-\eta \operatorname{grad} \mathcal{J}(\phi)\big)$$

*lies on the unique FR geodesic* $\gamma$ *starting at* $\phi$ *with initial velocity* $\dot\gamma(0) = -\eta \operatorname{grad} \mathcal{J}(\phi)$*; i.e.,* $\phi^+ = \gamma(1)$.

See proof in Appendix F.

**Corollary 1** (First-order equivalence to the additive update). *Let* $\mathcal{R}_\phi$ *be any retraction on* $\mathcal{M}$ *satisfying* $\mathcal{R}_\phi(0) = \phi$ *and* $D\mathcal{R}_\phi(0) = \operatorname{Id}$ *(the exponential map is a canonical retraction). In local coordinates,*

$$\operatorname{Exp}_\phi\big(-\eta F_\phi^{-1} \nabla_\phi \mathcal{J}\big) = \phi - \eta F_\phi^{-1} \nabla_\phi \mathcal{J} + O(\eta^2).$$

*By Proposition 1,* $\operatorname{grad} \mathcal{J} = F_\phi^{-1} \nabla_\phi \mathcal{J}$*. Hence the common additive natural-gradient step* $\phi^+ = \phi - \eta F_\phi^{-1} \nabla_\phi \mathcal{J}$ *is a first-order approximation to the geodesic update.*

**Natural Gradient on the Decoder.** For the decoder, we use the natural gradient on $\{p_\theta(y \mid z)\}_\theta$:

$$\theta_{t+1} = \theta_t - \eta_\theta F_\theta^{-1} \nabla_\theta \mathbb{E}_{x,y,z}[-\log p_\theta(y \mid z)], \tag{24}$$

where $F_\theta = \mathbb{E}_{q_\phi(z)}\mathbb{E}_{p_\theta(y|z)}\big[\nabla_\theta \log p_\theta(y \mid z) \nabla_\theta \log p_\theta(y \mid z)^\top\big]$. In practice, we compute $F_\theta^{-1} g$ using scalable approximations and solvers, such as K-FAC (Martens & Grosse, 2015; Martens et al., 2018). As the page limitation, we present the whole G-IB algorithm in Appendix G.

## 4 EXPERIMENTS

In this section, we conduct experiments to answer the following research questions (RQ) about G-IB:

- **RQ1**: How does the proposed G-IB perform on information compression and prediction accuracy, as compared with the state-of-the-art IB solutions? (See Sections 4.2 and 4.3)
- **RQ2**: How do different hyperparameters, such as the Lagrange multiplier $\beta$ and representation dimensionality $K$, influence the G-IB? (See Sections 4.4 and 4.5)

### 4.1 EXPERIMENTAL SETTINGS

**Datasets.** We have conducted experiments on three widely adopted public datasets: MNIST (Deng, 2012), CIFAR10 (Krizhevsky et al., 2009), and CelebA (Liu et al., 2018), offering a range of objective categories with varying levels of learning complexity. We present detailed statistics of all datasets and how do we use them in Appendix H.

**Models.** We select three model architectures of different sizes in our experiments: a 7-layer convolutional neural network (CNN), a 5-layer multi-layer perceptron (MLP), and ResNet18. For the MNIST dataset, we employ two MLPs to form the G-IB model (one as the compression encoder and one as the task decoder). For CIFAR10 and CelebA, we employ the ResNet18 as the encoder and one MLP and CNN as decoder for CIFAR10 and CelebA respectively.

**Metrics.** We quantify model utility by top-1 classification **Accuracy** on the held-out test set. Information compression is assessed by the mutual information *I(Z;X)* between the learned representation and the input, estimated with MINE (Belghazi et al., 2018). To probe the leakage contained in $Z$, we perform two standard representation-level attacks: (i) a model inversion attack that reconstructs $x$ from $z$ (Fredrikson et al., 2015), evaluated by mean squared error (**MSE**; lower MSE indicates stronger leakage); and (ii) a membership inference attack (Shokri et al., 2017), evaluated by the membership inference accuracy (**MIA**) (higher values indicate stronger leakage).

Table 1: General Evaluation Results on image datasets, MNIST and CIFAR10, and CelebA. Results in bold are the best; those in italics are the second best.

| Methods | MNIST, $\beta = 0.0001$, $K = 128$ | | | CIFAR10, $\beta = 0.0001$, $K = 128$ | | | CelebA, $\beta = 0.0001$, $K = 128$ | | |
|---|---|---|---|---|---|---|---|---|---|
| | Accuracy | $I(X; Z)$ | MSE | Accuracy | $I(X; Z)$ | MSE | Accuracy | $I(X; Z)$ | MSE |
| VIB (Alemi et al., 2016) | 98.72% | 1.81 | 0.034 | *82.65%* | 0.87 | *0.887* | 95.85% | 0.55 | 0.054 |
| SIB (Yang et al., 2025) | *99.08%* | 1.86 | 0.037 | 75.81% | **0.63** | 0.051 | **97.25%** | **0.38** | **0.072** |
| MINE (Belghazi et al., 2018) | 98.85% | 1.76 | 0.032 | 82.64% | 0.88 | 0.806 | 96.29% | 0.51 | 0.048 |
| AIB (Zhai & Zhang, 2022) | 99.01% | *1.69* | *0.043* | 82.63% | *0.84* | 0.812 | 96.05% | 0.52 | 0.053 |
| G-IB (Our) | **99.28%** | **1.69** | **0.043** | **85.54%** | 1.01 | **0.899** | *97.01%* | *0.47* | *0.066* |

**Compared IB Benchmarks.** We compare G-IB against four representative Information Bottleneck (IB) variants: (1) the standard Variational IB (**VIB**) (Alemi et al., 2016); (2) an IB variant where the mutual information term is estimated with **MINE** (Belghazi et al., 2018); (3) the state-of-the-art Structured IB (**SIB**) focusing on structure-aware feature learning (Yang et al., 2025); and (4) the Adversarial IB (**AIB**) that incorporates adversarial regularization into the bottleneck (Zhai & Zhang, 2022). For fairness, all methods use the same backbone, data preprocessing, and training schedule; hyperparameters are tuned on the validation set following the original papers where applicable.

## 4.2 OVERALL EVALUATION OF G-IB

**Setup.** We compare G-IB with four representative IB variants on MNIST, CIFAR10, and CelebA under the same backbone and schedule. We fix the Lagrange multiplier $\beta = 10^{-4}$ and the representation dimensionality $K = 128$ for an apples-to-apples comparison. We report top-1 accuracy (higher is better), the estimated mutual information $I(X; Z)$ via MINE (lower $I(X; Z)$ value indicates stronger compression), and model-inversion MSE from reconstructing $x$ from $z$ (higher indicates less leakage) in Table 1.

**Results.** Across the three datasets, G-IB attains the best or second-best results on all metrics. On MNIST, G-IB achieves the highest accuracy and a (tied) lowest $I(X; Z)$, while matching the top inversion MSE. This suggests strong compression without sacrificing utility. On CIFAR10, G-IB delivers the best accuracy and the highest inversion MSE (least leakage). Although SIB attains the lowest $I(X; Z)$, G-IB offers a better accuracy–privacy trade-off overall. On CelebA, SIB slightly leads in accuracy and MI. G-IB ranks second with competitive accuracy and privacy (indicated by MSE), confirming robustness on a more fine-grained, structured dataset.

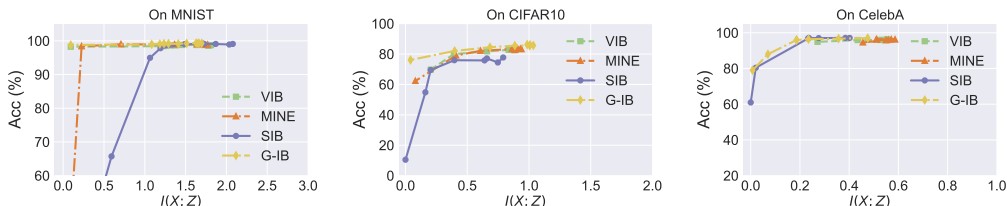

Figure 2: Evaluation of compression ratio and prediction accuracy from the information plane.

## 4.3 INFORMATION-PLANE EVALUATION: COMPRESSION VS. ACCURACY

**Setup.** We trace each method's information plane on MNIST, CIFAR10, and CelebA by randomly selecting the Lagrange multiplier $\beta$ from $10^{-6}$ to $10^1$. During this experiment, we fixed representation size $K = 128$. For every $\beta$, we train to convergence and record test accuracy (higher is better) and the estimated mutual information $I(X; Z)$ via MINE as shown in Figure 2.

**Results.** In Figure 2, across datasets, G-IB's curve is consistently shifted up and left relative to the baselines, achieving higher accuracy at a matched $I(X; Z)$, or a smaller $I(X; Z)$ at a matched accuracy (Pareto improvement). When compression is weak, i.e., large $I(X; Z)$, all methods reach a high-accuracy plateau. As compression strengthens, i.e., smaller $I(X; Z)$, VIB, MINE, and SIB exhibit clear accuracy drops, while G-IB maintains accuracy over a wider low-$I(X; Z)$ range before degrading. On CIFAR10 the gap is most visible, where G-IB preserves accuracy at lower $I(X; Z)$, And on CelebA, the curves cluster near the top but G-IB attains comparable accuracy with less information in $Z$.

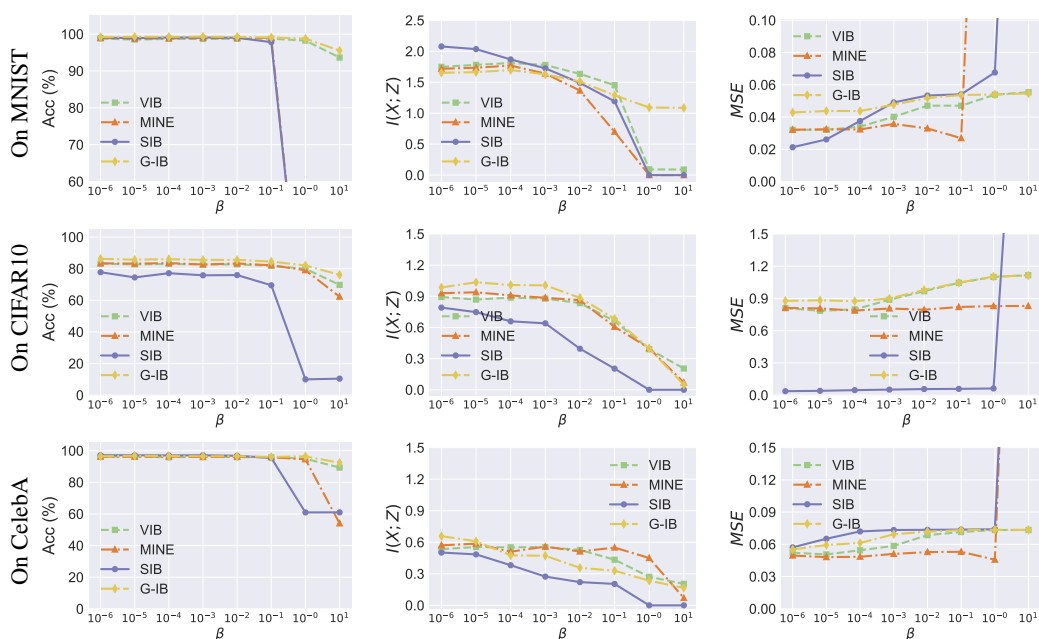

Figure 3: Evaluation about the impact of the Bottleneck multiplier $\beta$.

### 4.4 ABLATION: INFLUENCE OF THE BOTTLENECK MULTIPLIER $\beta$

**Setup.** We study the impact of the Bottleneck multiplier by sweeping $\beta$ on a logarithmic grid from $10^{-6}$ to $10^1$, with representation size fixed at $K = 128$. For each $\beta$, we train a model and report test accuracy (higher is better), the estimated mutual information $I(X; Z)$ (lower means stronger compression), and the model-inversion MSE from reconstructing $x$ from $z$ (higher means less leakage). Results for MNIST, CIFAR10, and CelebA are shown in Figure 3.

**Results.** (Left column) *Accuracy vs. $\beta$.* G-IB maintains the highest or near-highest accuracy over a wide range of $\beta$ on all datasets. Baselines, especially MINE and SIB, exhibit sharp degradation when $\beta \geq 10^{-1}$; on CIFAR10 and CelebA, SIB collapses at large $\beta$.

(Middle) $I(X; Z)$ *vs. $\beta$.* All methods show the expected monotonic decrease of $I(X; Z)$ as $\beta$ increases. SIB achieves the smallest $I(X; Z)$ (strongest compression) but at the cost of the pronounced accuracy drop above, whereas G-IB attains competitive compression while preserving accuracy.

(Right) *MSE vs. $\beta$.* G-IB yields the best or second-best MSE across most $\beta$, indicating stronger resistance to inversion attacks. At extreme $\beta$ ($10^0 - 10^1$), the MSE of some baselines spikes due to representation collapse. Reconstructions become effectively random, which inflates MSE but coincides with poor utility.

**Qualitative visualization.** To examine how the bottleneck strength shapes the representation, we visualize a 2-D t-SNE of the representation $\mu(x) = \mathbb{E}[Z \mid x]$ for 10,000 MNIST test images at three values of $\beta$ (Figure 4); the CIFAR10 results are in Figure 6 in Appendix I. With a small Bottleneck multiplier ($\beta = 10^{-4}$), clusters are well separated and exhibit relatively large within-class spread, indicating that $Z$ retains fine-grained input details. At $\beta = 10^0$, representation clusters contract toward class-wise prototypes while remaining separable. At $\beta = 10^1$, representation embeddings concentrate along narrow arcs near the class centers, consistent with stronger compression and the accuracy drop observed in Figure 3.

### 4.5 ABLATION: INFLUENCE OF THE REPRESENTATION DIMENSIONALITY $K$

**Setup.** We vary the representation dimensionality $K$ from $2^1$ to $2^9$ with the Lagrange multiplier fixed at $\beta = 10^{-4}$. For each $K$, we report the accuracy, mutual information $I(X; Z)$, MSE, and MIA for different IB methods. Results for MNIST and CIFAR10 are shown in Figure 5.

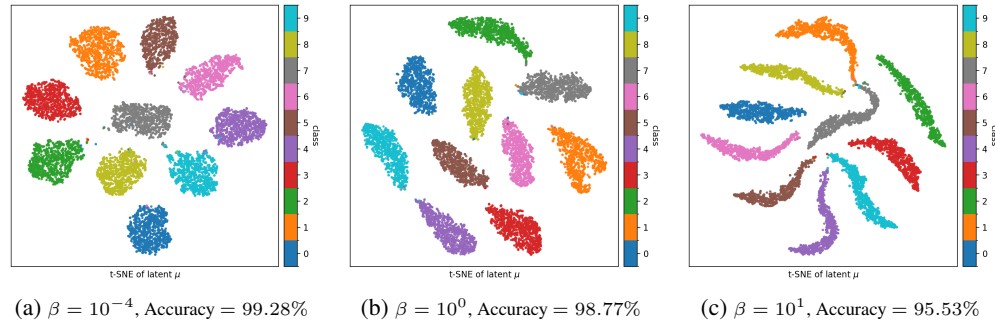

(a) $\beta = 10^{-4}$, Accuracy = 99.28%    (b) $\beta = 10^0$, Accuracy = 98.77%    (c) $\beta = 10^1$, Accuracy = 95.53%

Figure 4: Visualizing representation embeddings of posterior means $\mu(x)$ for 10,000 test images in two dimensions on MNIST ($K = 128$). Colors denote true labels. From left to right: $\beta = 10^{-4}, 10^0$, and $10^1$; the corresponding test accuracies are shown below each panel. As $\beta$ increases, within-class dispersion shrinks and clusters move toward class-wise prototypes, indicating stronger compression; accuracy decreases accordingly.

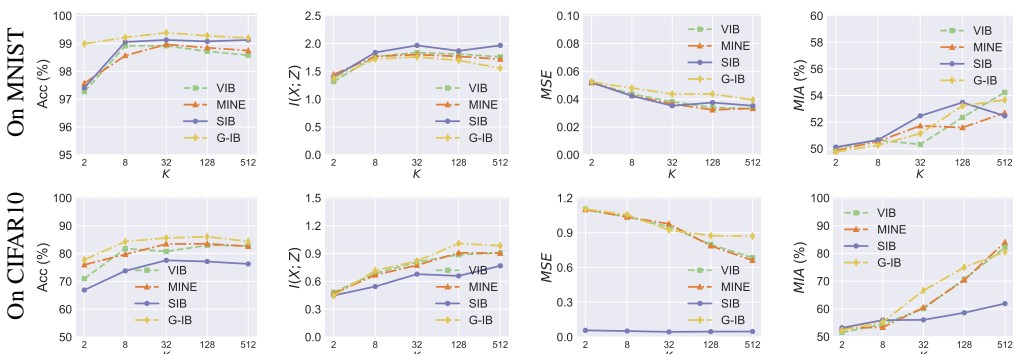

Figure 5: Evaluation about the impact of the Representation Dimensionality $K$.

**Results.** In the Left column of Figure 5, accuracy improves with $K$ and then saturates (MNIST: gains plateau around $K \geq 32$; CIFAR10: around $K \geq 128$). G-IB attains the highest or near-highest accuracy across $K$, especially in the low-to-mid range. In the Mid-left column, as expected, $I(X; Z)$ increases with $K$, indicating weaker compression when the latent space is wider. Tail non-monotonicity on MNIST at very large $K$ is minor and likely due to the representation dimensionality of $K = 32$ is already large enough for MNIST dataset. The Mid-right column shows that MSE generally decreases as $K$ grows (reconstructions become easier), reflecting increased leakage with higher-dimensional $Z$. G-IB keeps MSE competitively high (i.e., more resistant to inversion) at small and medium $K$. In the Right column, MIA success rises with $K$ on both datasets, corroborating that larger $Z$ carries more membership signal. Across a broad range of $K$, G-IB remains competitive; when $K$ is small-to-medium, it attains a favorable utility–privacy balance.

## 5 SUMMARY AND FUTURE WORK

We propose the G-IB to solve the IB problem from a information geometry perspective. The key ingredients are (i) a distribution-level Fisher–Rao (FR) discrepancy that locally matches KL to second order and is invariant under smooth reparameterizations of the latent, and (ii) a geometry-level Jacobian–Frobenius (JF) penalty that provides a local capacity-type upper bound on $I_\phi(Z; X)$ by discouraging pullback volume expansion. On the optimization side, we propose a natural gradient that aligned updates with the FR metric and prove that the standard additive natural-gradient step is first-order equivalent to the exponential-map (geodesic) update.

Promising directions include tightening the FR-based proxies beyond the local (second-order) regime and replacing the trace relaxation with sharper spectral approximations; connecting JF controlled smoothness to robustness and generalization bounds; and extending G-IB to federated and privacy-preserving settings where FR and JF controls might yield verifiable unlearning guarantees.

## STATEMENTS

**Ethic Statements.** Our study only involves the use of images of public datasets, with a long history of works having used these images for research. There are no new ethical implications for the humans within the datasets in this paper. Our work does not contain any user participants, and the outcome of this research influences benefits all individuals equally. As such "Respect for Persons" is satisfied.

**Reproducibility Statements.** To ensure the reproducibility of our research, we are committed to ensuring that our research is transparent, reproducible, and accessible to the broader community. The source code and the artifact of the G-IB is available at `https://anonymous.4open.science/r/G-IB-0569`.

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

## A  LLM USAGE DECLARATION

The authors declare that Large Language Models (LLMs) were used for grammar correction and text refinement. All research ideas, analyses, results, tables, and figures presented in this paper are original contributions by the authors and were not generated by LLMs.

## B  RELATED WORK

The Information Bottleneck (IB) Lagrangian (Tishby et al., 2000) has been widely studied in representation learning (Achille & Soatto, 2018; Rosati et al., 2024) and practical training techniques (Xu et al., 2022; Li et al., 2025b). A practical deep implementation is the Variational Information

Bottleneck (VIB) (Alemi et al., 2016), which introduces a variational encoder $q_\phi(z \mid x)$, a prior $r(z)$, and a decoder/classifier $q_\theta(y \mid z)$, and uses the following bounds:

$$
\begin{aligned}
I(Z;X) = \mathbb{E}_{p(x)}\big[\mathrm{KL}\big(q_\phi(z \mid x) \,\|\, q_\phi(z)\big)\big] &\leq \mathbb{E}_{p(x)}\big[\mathrm{KL}\big(q_\phi(z \mid x) \,\|\, r(z)\big)\big] \\
&= \mathbb{E}_{p(x)q_\phi(z|x)} \log q_\phi(z \mid x) \ - \ \mathbb{E}_{q_\phi(z)} \log r(z),
\end{aligned}
\tag{25}
$$

$$
I(Z;Y) = H(Y) - H(Y \mid Z) \ \geq \ H(Y) + \mathbb{E}_{p(x,y)\,q_\phi(z|x)} \log q_\theta(y \mid z).
\tag{26}
$$

Here $q_\phi(z) = \int q_\phi(z \mid x)p(x)\,dx$ is the encoder marginal. Eq. (25) uses $\mathrm{KL}\big(q_\phi(z)\,\|\,r(z)\big) \geq 0$ (with equality if $r(z) = q_\phi(z)$). Eq. (26) follows from the non-negativity of $\mathrm{KL}\big(p(y \mid z)\,\|\,q_\theta(y \mid z)\big)$ (cross-entropy bound).

Beyond VIB, a complementary line of work replaces the prior–KL surrogate with neural mutual-information estimators. These methods train a critic to optimize variational bounds on MI and related quantities, enabling more direct optimization of the IB objective. Representative examples include MINE (Belghazi et al., 2018), which maximizes a Donsker–Varadhan lower bound; NWJ and related $f$-divergence bounds (Letizia et al., 2024); and CLUB, which provides an upper bound useful for penalizing $I(Z;X)$ (Cheng et al., 2020). These estimators remove the need to choose a prior $r(z)$, but introduce practical trade-offs (bias/variance, saturation for large MI, dependence on negatives and critic capacity), as analyzed by McAllester & Stratos (2020).

Moreover, IB methods are studies across diverse scenarios. In graph learning, IB appears as the Graph Information Bottleneck (GIB) for minimal-sufficient node and structure representations that are also robust to perturbations (Wu et al., 2020b; Sun et al., 2022); follow-ups include variational GIB for subgraph recognition and VIB-guided graph structure learning that jointly optimizes topology and features (Yu et al., 2022). In semantic communication, IB gives a principled way to transmit meaning rather than raw bits. Task-oriented links use variational IB to trade informativeness vs. channel robustness and to handle distribution shift (Xie et al., 2023; Li et al., 2025a). The IB method also has informed the design of contrastive objectives and augmentation strategies, encouraging representations that discard nuisance variation yet retain label-sufficient information (Wei et al., 2022; Li et al., 2025b; Zhao et al., 2020; Xu et al., 2022).

Another important application scenario of IB is compressive privacy for privacy preserving. It achieves privacy protection of original data when participating in machine learning service via compressing the original data into low-dimensional space for the target machine learning task (Kung, 2018; Song et al., 2019; Zhang et al., 2023). Moreover, (Tseng & Wu, 2020) proposed a new privacy-preserving generative adversarial network (GAN) based on compressive pricacy. In their method (Tseng & Wu, 2020), the users will upload the compressed data $Z$ to the server side to achieve a machine learning service.

While effective, most of the above approaches hinge on explicit or variational MI estimates in high dimensions, which can be brittle. We exploit the *geometry* of the statistical manifold to implement information bottleneck. Specifically, in information geometry, the Fisher–Rao metric endows distributions with a Riemannian structure under which KL is locally the squared geodesic distance (Amari & Nagaoka, 2000); thus, a Fisher–Rao (FR) discrepancy $d_{\mathrm{FR}}^2\big(q_\phi(z \mid x), r(z)\big)$ provides a reparameterization-invariant surrogate for the $I(Z;X)$ compression term. Complementarily, viewing the encoder mean map $\mu_\phi : \mathcal{X} \to \mathcal{Z}$ as inducing a pullback metric $J_\mu^\top J_\mu$ suggests penalizing local volume distortion, which connects to contractive and Jacobian-based regularization (Ross & Doshi-Velez, 2018). We present the detailed introduction of the geometric information bottleneck in the methodology section.

APPENDIX

## C    PROOF OF EQUATION 3

*Proof.* Using the log-factorization identity and Fubini/Tonelli

$$\mathrm{KL}\big(p(x,z)\,\|\,q(x)r(z)\big) = \iint p(x,z)\,\log\frac{p(x,z)}{q(x)r(z)}\,dx\,dz$$

$$= \iint p(x,z)\,\log\frac{p(x,z)}{p_X(x)p_Z(z)}\,dx\,dz$$

$$+ \iint p(x,z)\,\log\frac{p_X(x)}{q(x)}\,dx\,dz + \iint p(x,z)\,\log\frac{p_Z(z)}{r(z)}\,dx\,dz$$

$$= \underbrace{\mathrm{KL}\big(p(x,z)\,\|\,p_X(x)p_Z(z)\big)}_{=\,I(X;Z)}$$

$$+ \underbrace{\int p_X(x)\,\log\frac{p_X(x)}{q(x)}\,dx}_{=\,\mathrm{KL}(p_X\|q)} + \underbrace{\int p_Z(z)\,\log\frac{p_Z(z)}{r(z)}\,dz}_{=\,\mathrm{KL}(p_Z\|r)}.$$

The last step uses Fubini's theorem to integrate out $z$ and $x$ respectively. Because each KL term is nonnegative, the minimum over $q, r$ is achieved at $q = p_X$, $r = p_Z$, with value $I(X; Z)$.    □

**Remark 1** (Information-geometric Pythagorean relation). *Since $\mathcal{I}$ is flat in the appropriate dual affine coordinates, the above decomposition is also the IG "Pythagorean theorem":*

$$\mathrm{KL}\big(p\,\|\,qr\big) = \mathrm{KL}\big(p\,\|\,p_Xp_Z\big) + \mathrm{KL}\big(p_Xp_Z\,\|\,qr\big), \qquad qr \in \mathcal{I}.$$

*Thus $p_Xp_Z$ is the e-projection of $p$ onto $\mathcal{I}$, and $I(X; Z)$ is exactly the projection distance.*

## D    PROOF OF LOCAL SECOND–ORDER (FR).

We show that for a regular parametric family $\{p_\theta : \theta \in \Theta \subset \mathbb{R}^d\}$ and $\theta'$ near $\theta$, i.e., $\Delta = \theta' - \theta$,

$$\mathrm{KL}\big(p_\theta\,\|\,p_{\theta'}\big) = \tfrac{1}{2}\,\Delta^\top F(\theta)\,\Delta \,+\, o(\|\Delta\|^2) = \tfrac{1}{2}\,d_{\mathrm{FR}}\big(p_\theta, p_{\theta'}\big)^2 \,+\, o(\|\Delta\|^2),$$

where $F(\theta)$ is the Fisher information and $d_{\mathrm{FR}}$ is the Fisher–Rao distance. Throughout we assume standard regularity: common support, differentiability up to second order in $\theta$, finiteness of $F(\theta)$, and interchange of expectation and differentiation.

Write

$$\mathrm{KL}\big(p_\theta\,\|\,p_{\theta'}\big) = \mathbb{E}_{p_\theta}[\log p_\theta(Z) - \log p_{\theta'}(Z)].$$

Fix $z$ and expand $\log p_{\theta'}(z)$ at $\theta$:

$$\log p_{\theta'}(z) = \log p_\theta(z) + \Delta^\top \nabla_\theta \log p_\theta(z) + \tfrac{1}{2}\,\Delta^\top \nabla_\theta^2 \log p_\theta(z)\,\Delta + o(\|\Delta\|^2).$$

Subtract from $\log p_\theta(z)$, take $\mathbb{E}_{p_\theta}$, and use $\mathbb{E}_{p_\theta}\big[\nabla_\theta \log p_\theta(Z)\big] = 0$ (zero mean score) to obtain

$$\mathrm{KL}\big(p_\theta\,\|\,p_{\theta'}\big) = -\tfrac{1}{2}\,\Delta^\top \mathbb{E}_{p_\theta}\big[\nabla_\theta^2 \log p_\theta(Z)\big]\,\Delta \,+\, o(\|\Delta\|^2).$$

By the information identity,

$$F(\theta) \;=\; \mathbb{E}_{p_\theta}\big[\nabla_\theta \log p_\theta(Z)\,\nabla_\theta \log p_\theta(Z)^\top\big] \;=\; -\,\mathbb{E}_{p_\theta}\big[\nabla_\theta^2 \log p_\theta(Z)\big],$$

hence

$$\mathrm{KL}\big(p_\theta\,\|\,p_{\theta'}\big) = \tfrac{1}{2}\,\Delta^\top F(\theta)\,\Delta \,+\, o(\|\Delta\|^2). \tag{27}$$

The FR metric is the Riemannian metric on $\Theta$ given by $g_\theta(u, v) = u^\top F(\theta)\,v$ for tangent vectors $u, v \in \mathbb{R}^d$. Let $\gamma : [0, 1] \to \Theta$ be any $C^1$ curve with $\gamma(0) = \theta$ and $\gamma(1) = \theta'$. Its FR length is

$$L(\gamma) \;=\; \int_0^1 \sqrt{\dot\gamma(t)^\top F(\gamma(t))\,\dot\gamma(t)}\,dt,$$

and the FR distance is $d_{\mathrm{FR}}(p_\theta, p_{\theta'}) = \inf_\gamma L(\gamma)$. For $\|\Delta\| \to 0$, choose the straight segment $\gamma(t) = \theta + t\Delta$ to get

$$L(\gamma)^2 = \Big( \int_0^1 \sqrt{\Delta^\top F(\theta + t\Delta)\,\Delta}\, dt \Big)^2 = \Delta^\top F(\theta)\,\Delta \,+\, o(\|\Delta\|^2),$$

using continuity of $F$ and a second–order expansion in $t$. Since the geodesic length is minimal,

$$d_{\mathrm{FR}}(p_\theta, p_{\theta'})^2 = \Delta^\top F(\theta)\,\Delta \,+\, o(\|\Delta\|^2). \tag{28}$$

This follows from standard Riemannian geometry: in normal coordinates at $\theta$, $d_{\mathrm{FR}}(p_\theta, p_{\theta'})^2 = \Delta^\top F(\theta)\Delta + O(\|\Delta\|^3)$, hence equation 28. Combining equation 27 and equation 28 yields the local equivalence

$$\mathrm{KL}(p_\theta \,\|\, p_{\theta'}) = \tfrac{1}{2}\, d_{FR}(p_\theta, p_{\theta'})^2 \,+\, o(\|\Delta\|^2).$$

Let $q_\phi(z|x)$ and a reference marginal $r(z)$ belong to $\{p_\theta\}$ with parameters $\theta(x)$ and $\theta_r$ respectively, and assume $\|\theta(x) - \theta_r\|$ is small for $p(x)$–almost every $x$. Applying the pointwise result above with $\Delta(x) = \theta(x) - \theta_r$ gives

$$\mathrm{KL}(q_\phi(z|x) \,\|\, r(z)) = \tfrac{1}{2}\, d_{\mathrm{FR}}(q_\phi(z|x), r(z))^2 \,+\, o(\|\Delta(x)\|^2).$$

Taking $\mathbb{E}_{p(x)}$ and using dominated convergence (guaranteed by the regularity assumptions) yields

$$\mathbb{E}_{p(x)}\mathrm{KL}(q_\phi(z|x) \,\|\, r(z)) = \tfrac{1}{2}\, \mathbb{E}_{p(x)}d_{\mathrm{FR}}(q_\phi(z|x), r(z))^2 \,+\, o\Big(\mathbb{E}_{p(x)}\|\Delta(x)\|^2\Big).$$

When $r(z) = p_\phi(z)$ (the aggregate posterior), this is precisely the local second–order approximation of the compression term $\mathbb{E}_{p(x)}\mathrm{KL}(q_\phi(z|x) \,\|\, p_\phi(z))$. □

## E    PROOF OF PROPOSITION 1

*Proof of Proposition 1.* By definition, the Riemannian gradient $\mathrm{grad}\,\mathcal{J}(\phi) \in T_\phi\mathcal{M} \simeq \mathbb{R}^d$ is the unique vector field satisfying, for all tangent directions $v \in T_\phi\mathcal{M}$,

$$\langle \mathrm{grad}\,\mathcal{J}(\phi),\, v\rangle_{g_\phi} \,=\, \mathrm{D}\mathcal{J}_\phi[v].$$

Under the Fisher–Rao metric, the inner product is $\langle u, v\rangle_{g_\phi} = u^\top F_\phi v$. In coordinates, the differential equals the Euclidean pairing with the usual gradient: $\mathrm{d}\mathcal{J}_\phi[v] = v^\top \nabla_\phi \mathcal{J}$. Hence, for all $v$,

$$v^\top F_\phi\, \mathrm{grad}\,\mathcal{J}(\phi) \,=\, v^\top \nabla_\phi \mathcal{J}.$$

Since $F_\phi$ is positive definite at regular points, we conclude $F_\phi\, \mathrm{grad}\,\mathcal{J}(\phi) = \nabla_\phi\mathcal{J}$, i.e. $\mathrm{grad}\,\mathcal{J}(\phi) = F_\phi^{-1}\nabla_\phi\mathcal{J}$. □

## F    PROOF OF THEOREM 1

*Proof of Theorem 1.* Let $v := -\eta\,\mathrm{grad}\,\mathcal{J}(\phi) \in T_\phi\mathcal{M}$. By existence and uniqueness for the geodesic equation with the Levi–Civita connection of the FR metric, there exists $\varepsilon > 0$ and a unique geodesic $\gamma_v : (-\varepsilon, \varepsilon) \to \mathcal{M}$ such that $\gamma_v(0) = \phi$ and $\dot{\gamma}_v(0) = v$.

The Riemannian exponential map at $\phi$ is defined by

$$\mathrm{Exp}_\phi(w) \,=\, \gamma_w(1) \quad \text{whenever 1 lies in the domain of } \gamma_w,$$

equivalently $\mathrm{Exp}_\phi(tw) = \gamma_w(t)$ for $t$ in a neighborhood of 0. (For global well-definedness one may restrict to $\|w\|$ below the injectivity radius.) Applying this with $w = v$ gives

$$\phi^+ \,=\, \mathrm{Exp}_\phi(v) \,=\, \gamma_v(1),$$

i.e., $\phi^+$ lies on the unique FR geodesic starting at $\phi$ with initial velocity $\dot{\gamma}(0) = v = -\eta\,\mathrm{grad}\,\mathcal{J}(\phi)$. □

---

**Algorithm 1:** G-IB Natural-Gradient Step (per iteration)

---

**Input:** Current params $(\phi_t, \theta_t)$; minibatch size $B$; step sizes $(\eta_\phi, \eta_\theta)$; bottleneck $\beta$; # Hutchinson probes $S$; damping $\lambda$; Fisher approx. mode K-FAC

**Output:** Updated params $(\phi_{t+1}, \theta_{t+1})$

1   **procedure GIB_Step** $(\phi_t, \theta_t, B, \eta_\phi, \eta_\theta, \beta, S, \lambda, \textit{mode})$ **:**

      `// 1) Sample minibatch and latent codes`

2      Draw minibatch $\{(x_i, y_i)\}_{i=1}^B$;;

3      $z_i \sim q_{\phi_t}(z \mid x_i)$ by reparameterization;

      `// 2) Estimate FR/JF terms (Hutchinson + JVPs)`

4      Estimate $\widehat{\mathcal{L}}_{\mathrm{FR}}$ and $\widehat{\mathcal{L}}_{\mathrm{JF}}$ using $S$ probe vectors and JVPs;

      `// 3) Compute Euclidean gradients`

5      $g_\theta \leftarrow \nabla_\theta \frac{1}{B} \sum_{i=1}^B \big[ -\log p_{\theta_t}(y_i \mid z_i) \big]$;

6      $g_\phi \leftarrow \nabla_\phi \left\{ \frac{1}{B} \sum_{i=1}^B \big[ -\log p_{\theta_t}(y_i \mid z_i) \big] + \beta\big(\widehat{\mathcal{L}}_{\mathrm{FR}} + \widehat{\mathcal{L}}_{\mathrm{JF}}\big) \right\}$;

      `// 4) Build Fisher approximations (Empirical Fisher or K-FAC)`

7      **if** *mode* = *K-FAC* **then**

8         Build layerwise Kronecker factors for encoder/decoder to obtain $\widehat{F}_\phi$ and $\widehat{F}_\theta$;

9      $\widehat{F}_\theta^\lambda \leftarrow \widehat{F}_\theta + \lambda I, \quad \widehat{F}_\phi^\lambda \leftarrow \widehat{F}_\phi + \lambda I$;

      `// 5) Solve for natural directions (no explicit inversion)`

10     Find $v_\theta$ s.t. $(\widehat{F}_\theta^\lambda) v_\theta = g_\theta$ via Conjugate Gradient (CG);

11     Find $v_\phi$ s.t. $(\widehat{F}_\phi^\lambda) v_\phi = g_\phi$ via CG;

      `// 6) Parameter updates (natural gradients)`

12     $\theta_{t+1} \leftarrow \theta_t - \eta_\theta \, v_\theta$          (Eq. (24));

13     $\phi_{t+1} \leftarrow \phi_t - \eta_\phi \, v_\phi$           (Eq. (21));

14     **return** $(\phi_{t+1}, \theta_{t+1})$;

---

# G   THE G-IB ALGORITHM

We provide the algorithm of G-IB as the following Algorithm 1.

Algorithm 1 executes one G-IB training iteration with *natural-gradient* updates for the decoder $\theta$ and encoder $\phi$ under a K-FAC curvature mode. (Step 1) Given a minibatch $\{(x_i, y_i)\}_{i=1}^B$, latent codes are drawn via the reparameterized posterior $z_i \sim q_{\phi_t}(z \mid x_i)$. (Step 2) The geometry-aware bottleneck surrogates are estimated without forming full Jacobians: the Fisher–Rao proxy $\widehat{\mathcal{L}}_{\mathrm{FR}}$ and the Jacobian–Frobenius penalty $\widehat{\mathcal{L}}_{\mathrm{JF}}$ are computed using $S$ Hutchinson probe vectors together with Jacobian–vector products (JVPs). (Step 3) We then compute the Euclidean gradients of the decoder NLL and the full encoder objective,

$$g_\theta = \nabla_\theta \frac{1}{B} \sum_{i=1}^B \big[ -\log p_{\theta_t}(y_i \mid z_i) \big], \qquad g_\phi = \nabla_\phi \left\{ \frac{1}{B} \sum_{i=1}^B \big[ -\log p_{\theta_t}(y_i \mid z_i) \big] + \beta\big(\widehat{\mathcal{L}}_{\mathrm{FR}} + \widehat{\mathcal{L}}_{\mathrm{JF}}\big) \right\}.$$

(Step 4) To obtain natural directions, we build *layerwise* Kronecker-factored Fisher approximations for both networks. For each layer $\ell$ with weight matrix $W_\ell$, K-FAC uses the block-diagonal model $F_\ell \approx A_\ell \otimes G_\ell$, where $A_\ell := \frac{1}{B} \sum_i a_{\ell,i} a_{\ell,i}^\top$ is the covariance of layer inputs (activations) and $G_\ell := \frac{1}{B} \sum_i g_{\ell,i} g_{\ell,i}^\top$ is the covariance of backpropagated pre-activation gradients. Tikhonov damping yields $\widehat{F}_\theta^\lambda = \widehat{F}_\theta + \lambda I$ and $\widehat{F}_\phi^\lambda = \widehat{F}_\phi + \lambda I$. (Step 5) Rather than inverting these matrices, we solve the linear systems $(\widehat{F}_\theta^\lambda) v_\theta = g_\theta$ and $(\widehat{F}_\phi^\lambda) v_\phi = g_\phi$ via conjugate gradients (CG). Each CG iteration only needs Fisher–vector products, which K-FAC supplies efficiently: if $V_\ell$ reshapes the vector $v$ to the layer's weight shape, then

$$\mathrm{FVP}_\ell(v) \;=\; \mathrm{vec}\Big( (G_\ell + \lambda I) \, V_\ell \, (A_\ell + \lambda I) \Big).$$

(Step 6) Finally, parameters are updated along the natural directions: $\theta_{t+1} = \theta_t - \eta_\theta v_\theta$ (cf. Eq. (24)) and $\phi_{t+1} = \phi_t - \eta_\phi v_\phi$ (cf. Eq. (21)). The hyperparameters $(\eta_\phi, \eta_\theta, \beta, S, \lambda)$ control step sizes,

compression strength, estimator variance, and conditioning, while K-FAC trades a faithful curvature signal for scalable, inversion-free natural-gradient steps.

Table 2: Dataset statistics.

| Dataset | Feature Dimension | #. Classes | #. Samples |
|---|---|---|---|
| MNIST (Deng, 2012) | 28×28×1 | 10 | 70,000 |
| CIFAR10 (Krizhevsky et al., 2009) | 32×32×3 | 10 | 60,000 |
| CelebA (Liu et al., 2018) | 178×218×3 | 2 (Gender) | 202,599 |

## H  DATASETS

The statistics of all datasets used in our experiments are listed in Table 2. Both MNIST and CIFAR10 are used to train 10-class classification models. The experiment on CelebA is to identify the gender attributes of the face images. The task is a binary classification problem, different from the ones on MNIST and CIFAR10. These datasets offer a range of objective categories with varying levels of learning complexity. We also introduce them as below.

- **MNIST (Deng, 2012).** MNIST contains 60,000 handwritten digit images for the training and 10,000 handwritten digit images for the testing. All these black and white digits are size normalized, and centered in a fixed-size image with $28 \times 28$ pixels.
- **CIFAR10 (Krizhevsky et al., 2009).** CIFAR10 dataset consists of 60,000 32x32 colour images in 10 classes, with 6,000 images per class. There are 50,000 training images and 10,000 test images.
- **CelebA (Liu et al., 2018).** CelebA is a large-scale face attributes dataset with more than 200,000 celebrity images, each with 40 attribute annotations.

## I  ADDITIONAL EXPERIMENTS

**Qualitative visualization.** To examine how the bottleneck strength shapes the representation, we visualize a 2-D t-SNE of the representation $\mu(x) = \mathbb{E}[Z \mid x]$ for 10,000 CIFAR10 test images at three values of $\beta$ (Figure 6). With a small Bottleneck multiplier ($\beta = 10^{-4}$), clusters are well separated and exhibit relatively large within-class spread, indicating that $Z$ retains fine-grained input details. At $\beta = 10^0$, representation clusters contract toward class-wise prototypes while remaining separable. At $\beta = 10^1$, representation embeddings concentrate along narrow arcs near the class centers, consistent with stronger compression and the accuracy drop observed in Figure 3.

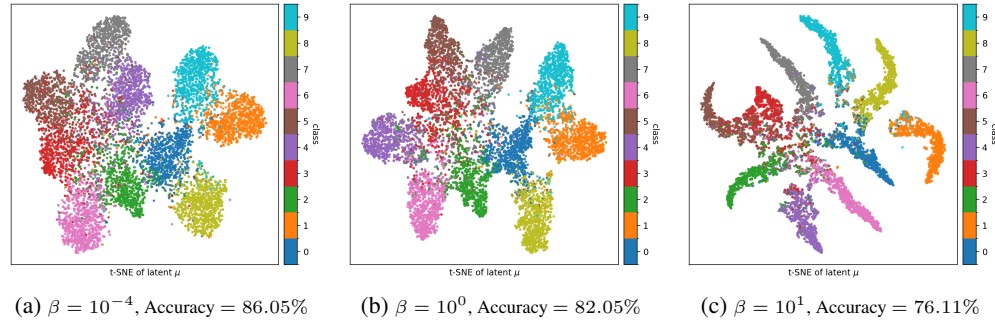

(a) $\beta = 10^{-4}$, Accuracy = 86.05%   (b) $\beta = 10^0$, Accuracy = 82.05%   (c) $\beta = 10^1$, Accuracy = 76.11%

Figure 6: Visualizing representation embeddings of 10000 test images in two dimensions on CIFAR10. The images are colored according to their true class label. We $\beta$ becomes larger, we forget more about the input and the representation embedding of each class is compressed close to the average $\mu$. We also report the test accuracy, which decreases as $\beta$ increases.

