# OpenReview forum: "Geometric IB: Improving Information Bottleneck with Geometry-Aware Compression on Statistical Manifolds"
_ICLR.cc/2026/Conference — Submitted to ICLR 2026_

### Official Review · Reviewer_E6Kx · 2025-10-28

**Soundness:** 2
**Presentation:** 2
**Contribution:** 3
**Rating:** 2
**Confidence:** 4

**Summary:**

The paper proposes Geometric Information Bottleneck (G-IB), which reinterprets the Information Bottleneck (IB) framework through the lens of information geometry. The authors claim to avoid direct mutual information estimation by introducing two geometry-aware regularizations. G-IB achieves a better trade-off between prediction accuracy and compression ratio.

**Strengths:**

1. The idea of combining geometric considerations (via Fisher–Rao metrics) with IB is interesting and theoretically motivated.

**Weaknesses:**

1. Extra computation for surrogates: computing JF uses Hutchinson probes + JVPs (adds forward-mode calls per sample). FR requires either closed-form KL or the FR distance proxy.

2. Heavier optimizer: They propose natural-gradient updates via K-FAC with CG solves (Algorithm 1), which is more involved than standard Adam/SGD. No comparison is made with standard optimizers like Adam or SGD to demonstrate practical benefit.

3. The reported accuracy gains (Table 1) are minor and inconsistent. For instance, on CelebA, G-IB underperforms compared to SIB, and the improvements on MNIST and CIFAR-10 are negligible.

4. All datasets used (MNIST, CIFAR10, CelebA) are low-complexity and saturated benchmarks. There is no evidence that the proposed method scales to modern architectures or large-scale data (e.g., ImageNet).

**Questions:**

1. Please include experiments on more challenging datasets such as CIFAR-100 and Tiny-ImageNet to better demonstrate the scalability and generalization ability of the proposed method. In addition, an evaluation of adversarial robustness would significantly strengthen the empirical claims.

2. The paper should discuss and compare with several closely related approaches, including NIB (Nonlinear Information Bottleneck) [1], DIB [2] (Deep Deterministic Information Bottleneck), and HSIC-Bottleneck [3], which are all relevant to the proposed framework.

[1] Kolchinsky, Artemy, Brendan D. Tracey, and David H. Wolpert. "Nonlinear information bottleneck." Entropy 21.12 (2019): 1181.

[2] Yu, Xi, et,al. "Deep deterministic information bottleneck with matrix-based entropy functional." ICASSP 2021-2021 IEEE International Conference on Acoustics, Speech and Signal Processing (ICASSP). IEEE, 2021.

[3] Ma, Wan-Duo Kurt, J. P. Lewis, and W. Bastiaan Kleijn. "The HSIC bottleneck: Deep learning without back-propagation." Proceedings of the AAAI conference on artificial intelligence. Vol. 34. No. 04. 2020.

---

### Official Review · Reviewer_iDY5 · 2025-10-28

**Soundness:** 3
**Presentation:** 3
**Contribution:** 2
**Rating:** 4
**Confidence:** 2

**Summary:**

The authors revisit the Information Bottleneck (IB) through information geometry. They show that both information terms can be written as minimal KL distances from the joint to independence sub-manifolds (“Pythagorean” projection view). Building on this, they propose Geometric IB (G‑IB), which avoids explicit MI estimation at training time and instead controls compression with two terms; Fisher–Rao discrepancy and Jacobian–Frobenius. G‑IB is compare to VIB, MINE‑based IB, a structured IB (SIB), and adversarial IB (AIB) on several classic vision datasets, demonstrating G-IBs consistent performance.

**Strengths:**

- **Originality**: the Geometric IB paper's primary contribution is the explicit synthesis and formalization of these concepts into a coherent, geometry-aware framework specifically designed to solve the Information Bottleneck problem.

- **Quality**: the work presents strong theoretical foundation and accompanied by empirical validation. The reformulation of MI as a geometric projection provides a principled basis for the rest of the work. The derivations of the geometric penalties are well-motivated, and connected to the suggested optimization setup. Experiments span relevant baselines (VIB, MINE, SIB, AIB) on standard benchmarks (MNIST, CIFAR10, CelebA), and provide desired analysis for the IB setting -- the information plane.

- **Clarity**: the paper construction is clear and easy to follow, the problem is motivated and the theoretical background/setting is provided in a an understoodable manner -- even for non-expert readers.

- **Significance**: training without direct MI estimation alleviates known issues for IB based optimization.

**Weaknesses:**

- **Scalability**: the authors claim for scalability, however computation is still quite expensive-- actual applicable scales are not presneted.
- **Contributions**: Experiments are conducted only on classical vision datasets, presenting minimal improvement.
- **Geometric motivation**: following the previous point, geometric aspects / problems are not explored within provided experiments.

**Questions:**

Following the above weaknesses, could the authors:
1. elaborate on scalability;
2. extend the experiments to additional cases better representing the contribution and potentially highlighting the geometric aware setting.

---

### Official Review · Reviewer_saLk · 2025-10-29

**Soundness:** 2
**Presentation:** 3
**Contribution:** 2
**Rating:** 4
**Confidence:** 4

**Summary:**

The authors propose the Geometric Information Bottleneck (G-IB), a revised version of the well-known Information Bottleneck principle. G-IB controls information compression using two additional terms based on geometric quantities. According to the authors, G-IB achieves a better trade-off between prediction accuracy and compression ratio in the information plane compared to the mainstream IB method (MINE).

**Strengths:**

Exploring the Information Bottleneck phenomenon from an information geometry perspective appears interesting and promising. Key components of the proposed approach include: 1) a geometry-level Jacobian–Frobenius (JF) penalty that provides a local capacity-type upper bound on $I_{\phi}(X, Z)$ by discouraging pullback volume expansion; 2) a distribution-level Fisher–Rao (FR) discrepancy that locally matches KL divergence to second order and is invariant under smooth reparameterizations of the latent variable. The geometric perspective on mutual information represents a promising research direction in information theory. However, the paper has several limitations and concerns, detailed below.

**Weaknesses:**

It would be valuable to conduct additional experiments exploring the dependencies between $I(X, Y)$ and $I(X, Z)$ during neural network training and to provide the resulting information planes, similar to those in (Shwartz-Ziv \& Tishby, 2017).

The use of the Fisher-Rao metric in a natural Riemannian gradient descent method was first introduced in [1]. Proposition 1 (lines 254–260) in this paper appears to restate Theorem 1 from [1]. Moreover, [2] contains theoretical results that strongly intersect with those presented here: for instance, the proof of local second-order Fisher–Rao approximation (Appendix D) resembles Section 2.1.2 of [2], and Proposition 2 (lines 263–269) is similar to Proposition 1 in [2] (latter part of Section 2.1.2). Overall, [1, 2] already establish the relationship between the Fisher–Rao metric and Kullback–Leibler divergence from the perspective of “shortest path uphill.”

I also have concerns regarding the experimental pipeline. How do the authors verify that the estimated value of $I(X, Z)$ is sufficiently accurate relative to the true mutual information? Table 1 compares several IB benchmarks but does not address the accuracy of MI estimation. It is important to demonstrate that the estimated MI values are good approximations of the true values. I recommend that the authors include experiments on synthetic datasets to provide empirical justification for their approach.

Additionally, in lines 480–482, the authors state: “we propose a natural gradient that aligns updates with the FR metric and prove that the standard additive natural-gradient step is first-order equivalent to the exponential-map (geodesic) update.” However, similar results have been established in prior works, such as [1, 3].

Furthermore, many MI estimators have been proposed since MINE that should be included in the benchmarking. For example, [4–7] and references therein. To ensure a fair comparison, some recent MI estimation methods should be added to the experimental section. The paper would also benefit from more real-world experiments and a broader discussion of the method’s practical advantages (e.g., scalability and data efficiency).

Minor issues:

 - The definition of Fisher information $F(\theta)$ should be added to the main text (lines 161–162).
 - The quality of Figures 2–5 could be improved (e.g., increase font sizes of axis labels).

References:

[1] S. Amari. Natural Gradient Works Efficiently in Learning. Neural Computation, 10(2):251–276, 1998.

[2] Y. Ollivier, L. Arnold et al. Information-Geometric Optimization Algorithms: A Unifying Picture via Invariance Principles. J. Machine Learning Research, 18(18):1–65, 2017.

[3] S. Amari, H. Nagaoka. Methods of Information Geometry, 2000.

[4] T. Nishiyama. A New Lower Bound for Kullback–Leibler Divergence Based on Hammersley–Chapman–Robbins Bound, 2019.

[5] I. Butakov, A. Tolmachev et al. Information Bottleneck Analysis of Deep Neural Networks via Lossy Compression. ICLR, 2024.

[6] I. Butakov, A. Tolmachev et al. Mutual Information Estimation via Normalizing Flows. NeurIPS, 2024.

[7] G. Franzese, M. Bounoua, P. Michiardi. MINDE: Mutual Information Neural Diffusion Estimation. ICLR, 2024.

**Questions:**

1. How do you estimate the convergence rate of your method? Could you comment on the precision of the approximations suggested in Equations (6) and (11)?
2. Why is $S = 1$ or $S = 2$ (line 220) typically sufficient for good trace estimation?
3. Why are [1] (Amari, 1998) and [2] not cited, despite the significant overlap in theoretical content? Could you clearly highlight your contributions and theoretical novelty?
4. How do you verify the quality of MI estimation using your method? Why was MINE chosen over more recent MI estimation approaches?
5. What is the relationship between minimizing the Fisher–Rao and Jacobian–Frobenius terms in Equation (19)? How do they jointly affect the minimization of the overall G-IB objective?

---

### Meta-Review · Area_Chair_nGZh · 2026-01-05

**Summary:**

Reviewer saLk: Raises major novelty concerns, noting that several core theoretical results closely parallel prior work ([1–3]). Also questions the experimental validity of the MI estimation, asking for synthetic‐data validation and broader benchmarking with more recent MI estimators.

Reviewer iDY5: Questions scalability and practical relevance, noting that computation remains expensive and improvements are minimal.

Reviewer E6Kx: Highlights substantial computational overhead, lack of comparison with standard optimizers, minor/inconsistent performance gains, and limited evaluation on only small, saturated benchmarks, with no evidence of scalability to modern large‐scale settings.

**Reviewer Concerns:**

The authors did not participate in the rebuttal.

**Reviewer Scores:**

Reviewer saLk: 4
Reviewer iDY5: 4
Reviewer E6Kx: 2

As no rebuttal was submitted by the authors, I expect the reviewers will maintain their original scores.

---

### Decision · Program_Chairs · 2026-01-26

Reject